# Neutron Shielding Performance of 3D-Printed Boron Carbide PEEK Composites

**DOI:** 10.3390/ma13102314

**Published:** 2020-05-18

**Authors:** Yin Wu, Yi Cao, Ying Wu, Dichen Li

**Affiliations:** 1School of Aerospace Engineering, Xi’an Jiaotong University, Xi’an 710049, China; wyxjtu@stu.xjtu.edu.cn; 2State Key Laboratory for Manufacturing Systems Engineering, Xi’an Jiaotong University, Xi’an 710049, China; xjtudcli@sina.com; 3Nuclear Power Institute of China, Chengdu 610213, China

**Keywords:** FDM, PEEK composites, neutron shielding, boron carbide

## Abstract

Polyethylene is used as a traditional shielding material in the nuclear industry, but still suffers from low softening point, poor mechanical properties, and difficult machining. In this study, novel boron carbide polyether-ether-ketone (PEEK) composites with different mass ratios were prepared and tested as fast neutron absorbers. Next, shielding test pieces with low porosity were rapidly manufactured through the fused deposition modeling (FDM)-3D printing optimization process. The respective heat resistances, mechanical properties, and neutron shielding characteristics of as-obtained PEEK and boron carbide PEEK composites with different thicknesses were then evaluated. At load of 0.45 MPa, the heat deformation temperature of boron carbide PEEK increased with the boron carbide content. The heat deformation temperature of 30% wt. boron carbide PEEK was recorded as 308.4 °C. After heat treatment, both tensile strength and flexural strength of PEEK and PEEK composites rose by 40%–50% and 65%–78%, respectively. Moreover, the as-prepared composites showed excellent fast neutron shielding performances. For shielding test pieces with thicknesses between 40 mm and 100 mm, the neutron shielding rates exhibited exponential variation as a function of boron carbide content. The addition of 5%–15% boron carbide significantly changed the curvature of the shielding rate curve, suggesting an optimal amount of boron carbide. Meanwhile, the integrated shielding/structure may effectively shield neutron radiation, thereby ensuring optimal shielding performances. In sum, further optimization of the proposed process could achieve lightweight materials with less consumables and small volume.

## 1. Introduction

With the increasing use of nuclear energy in various sectors [1], growing concerns about radiation safety and protection issues are being considered [2,3,4]. Neutron radiation generated by nuclear reactors and accelerators poses a huge threat to humans and electronic components as it can strongly penetrate biological bodies and electronic devices [5]. Traditional neutron shielding materials include concrete and water, in which concrete possesses better shielding properties [6,7,8], but is not convenient for moving from one spot to another. By comparison, water contains H, which makes it an excellent neutron attenuation body, but might negatively impact the overall mechanical properties. Moreover, nuclear reactors with miniaturized sizes and lightweight fashion have been developed [9]. However, smaller reactors require much higher levels of nuclear fuel [10], leading to much more released heat during combustion [11]. Hence, the development of nuclear radiation protection materials with improved shielding protection, but limited volume, structure, and high-temperature environment has become an important and urgent task.

As one promising and significant ceramic material [12], boron carbide stands out with its extraordinary properties [13,14], such as ultra-high hardness (Mohs hardness of 9.36 and microhardness of 55 GPa to 67 GPa) [15], less density (2.52 g/m^3^), high melting point (2350 °C), high boiling point (3500 °C), high temperature strength, as well as good chemical stability without reacting with strong acids or alkali solutions. In addition, boron carbide has been widely applied in the national defense industry [16,17], nuclear industry, and other fields [18,19,20], on account of its high chemical degree, neutron absorption, wear resistance, and excellent semiconductor conductivity. In view of its outstanding properties, boron carbide is usually used to produce bullet-proof materials; nozzles for guns and cannons; and the essential components of nuclear reactors, such as the control rods, accident rods, safety rods, plates, or neutron absorbers [21]. Furthermore, combined with cement [22], it is also used for the nuclear shielding body, which is the most important functional element second only to the nuclear fuel elements. Boron is an excellent neutron absorber with high neutron capture cross section [23,24,25,26], excellent corrosion resistance, and no produced radioisotopes. Boron polymers are effective neutron shielding materials [27,28,29], thereby widely used in control rods of reactors and rocket fuel. In particular, a high boron content makes boron carbides important neutron-absorbing materials in the nuclear industry [30].

Light elements like H could effectively shield fast neutron radiation thanks to their scattering power. Thus, polyethylene with a high H content has long been utilized as a neutron shielding material [31]. With a melting point of 130 °C and a relative density of 0.941 to 0.960 g/cm^3^, polyethylene possesses good chemical stability and good resistance to heat, cold, and environmental stress cracking. Polyethylene molds around 140 to 220 °C, and possesses excellent corrosion resistance and electrical insulation. In addition, with less water absorption, polyethylene also has good electrical properties and radiation resistance [32,33,34,35,36]. However, as polyethylene is pressure-sensitive, injection under high pressure during molding is needed [37]. In the process of injection, it is crucial to ensure the uniformity of material temperature. In the meantime, the heating time should not be too long, for long-time hearing may give rise to the decomposition of polyethylene. Compared with other plastic materials, polyethylene is relatively soft, lightweight, and transparent. More importantly, polyethylene is non-toxic [38], so it is harmless to the human body. Although the mechanical properties of polyethylene are general, such as low tensile strength and poor creep resistance, it demonstrates good impact resistance [39]. Polyethylene can be mainly divided into linear low-density polyethylene (LLDPE), low-density polyethylene (LDPE), and high-density polyethylene (HDPE). In terms of impact strength, LDPE > LLDPE > HDPE, and in terms of mechanical properties, LDPE < LLDPE < HDPE. Thereby, HDPE has excellent physical and mechanical properties, but it is difficult to be processed in extruder owing to its long and tangled molecular chain; extremely high melt viscosity [40,41]; decomposition after heating; and extreme insensitivity to thermal shear stress, which may result in shear fracture. On the other hand, boron carbide (B_4_C) polyethylene (PE) composite has demonstrated excellent performance in shielding neutron radiation owing to the presence of high H content in polyethylene [42,43,44]. However, polyethylene is difficult to use in real applications owing to its low melting point, weak heat resistance (softening at 110 °C), weak mechanical strength, and poor radiation corrosion resistance [45]. Neutron shielding requires particularly high-temperature resistant polymer matrices, where epoxy resin or rubber have been used as matrices, but cannot endure temperature above 170 °C [46]. Therefore, the development of novel high-temperature-resistant materials with improved neutron shielding properties is highly desirable. With 143 °C glass transition temperature (Tg) and 334 °C melting point, polyether-ether-ketone (PEEK) can achieve 48% maximum crystallinity [47,48,49]. The density in amorphous state is 1.265 g/cm^3^, while in maximum crystallinity state, its density is 1.32 g/cm^3^. Thanks to its low density, PEEK is adopted instead of steel to achieve light weight in aerospace [50]. Moreover, its crystal structure means it possesses excellent heat resistance and mechanical properties [51,52,53,54]. It can be used at 250 °C for a long time, and its instantaneous temperature can reach 300 °C, and even at 400 °C, it can endure a short period without decomposition. Compared with other high temperature resistant plastics [55,56], such as polyimide (PI), polyphenylene sulfide (PPS), polytetrafluoroethylene (PTFE), and polyphenylene oxide (PPO), the upper temperature limit of PEEK is nearly 50 °C higher than that of other plastics. With great rigidity, good dimensional stability, and small linear expansion coefficient, PEEK is very close to aluminum. With excellent corrosion resistance similar to nickel steel [57], PEEK can only be dissolved or destroyed by concentrated hydrochloric acid. It is also flame-retardant, and even in flame conditions, it releases less smoke and toxic gases. With good radiation resistance, it possesses the best fatigue of alternating stress among all plastics [58,59,60,61], which is close to alloy materials. Most significantly, high wear resistance and low coefficient of friction at 250 °C enable PEEK to be used as a military material in the manufacture of various aircraft parts for aerospace applications, for PEEK can largely replace other metal materials such as aluminum. Among electromagnetic radiation resistant aromatic hydrocarbon polymers without halogen elements, PEEK has been shown to be useful as insulating cable material with excellent mechanical properties [62], good thermal and chemical stability, and superior abrasion resistance. However, its high melting point and low melting index render its processing more challenging and expensive. So far, PEEK and its composites have widely been used in aerospace, electronics, and power and medical equipment [63,64,65,66,67]. However, their usage in nuclear shielding is limited, partly owing to their high cost and difficult processing [68].

For instance, Okuno et al. processed shielding protection materials using two methods [69]. The first consisted of high-temperature injection mold and the second was based on film pressure on thin layer resin films. For materials with complex structures and low melting indexes, shark skin disease could be induced on surfaces of bulk of processed products, often caused by frequent melt fracture during processing [70]. Furthermore, the whole processing could last longer periods, and be characterized by defects and difficulties.

Recently, 3D printing (additive manufacturing) based on fused deposition modeling (FDM) has gained worldwide popularity in various industries owing to its rapid manufacturing [71,72,73], integrated design and manufacturing, high material utilization, and suitability for molding of complex microstructures [74]. However, no relevant literature dealing with FDM-based manufacturing and processing of nuclear protection equipment has so far been published.

In this paper, a new type of shielding PEEK composites was developed. Screw extruder was adopted for palletization and extrusion, and 3D printing (FDM) was used to manufacture protection test pieces with different sizes, contents, and structures. The heat deformation temperatures, mechanical properties, and shielding performances of the as-obtained composites were all evaluated and the results were discussed.

## 2. Experimental

### 2.1. Material Preparation

Boron carbide powder (Dunhua Zhengxing Abrasive Co., Ltd., Jilin, China) at different weights (G50 = 1.2 µm) and polyether-ether-ketone (PEEK, VICTREX, 450PF, Lancashire, United Kingdom) were subjected to vacuum drying, mixing, air dispersion, screw granulation, and extrusion to yield boron carbide PEEK composites with different boron carbide contents (10%, 20%, and 30%). The test pieces were produced by FDM (Figure 1). 

### 2.2. Shielding Process

After preparation of boron carbide PEEK shielding materials with different contents, heat treatment was carried out in a nozzle to a semi-flow state for covering, which then covered the previous layers along the pre-designed path and filling mode. Next, rapid cooling and solidification were performed under air or water cooling, where the extruded wire was bonded to the surrounding materials. The test pieces were fabricated by self-developed high-temperature 3D printer (Figure 1c), assembled at the Xi’an Jiaotong University. To prevent deterioration in shielding performance, reduction in voids, or increase in void ratio, the whole 3D printing processing of pieces was explored.

During the shielding of test bodies, the final print quality would mainly be affected by the nozzle temperature, diameter of print nozzle, nozzle speed, width of filling line (line width), and overlapping ratio of the filing lines (line-to-line overlapping ratio). Figure 2a_1_, b_1_, and d_1_ used the same materials with 10% wt. boron carbide PEEK and Figure 2c_1_ used the materials with PEEK. Because the nozzle temperature was insufficient and the nozzle speed was too fast, a great number of voids were formed in the small filling line space (Figure 2a_1_). 

Besides, with large line-to-line overlapping ratios, the materials at line junction overflowed after melting (Figure 2b_1_). As the nozzle plane was affected by the interface, the increase in moving resistance and delay in movement trajectory may lead to misalignment (Figure 2b_1_). The qualified test pieces mean the test pieces that possess a smooth surface without many holes, which would better fit with Figure 2c_1_,d_1_. By comparison, unqualified test pieces were those presented in Figure 2a_1_,b_1_. The experimental specific processing parameters could be summarized as follows:In Figure 2a_1_: nozzle temperature = 410 °C, diameter of print nozzle = 0.4 mm, nozzle speed = 80 mm/s, line width = 0.4 mm, and line-to-line overlapping ratio = 0.In Figure 2b_1_: nozzle temperature = 425 °C, diameter of print nozzle=0.4 mm, nozzle speed = 20 mm/s, line width = 0.4 mm, and line-to-line overlapping ratio = 50%.In Figure 2c_1_: nozzle temperature = 410 °C, diameter of print nozzle = 0.4 mm, nozzle speed = 20 mm/s, line width = 0.4 mm, and line-to-line overlapping ratio = 10%.In Figure 2d_1_: nozzle temperature = 425 °C, diameter of print nozzle = 0.4mm, nozzle speed = 20 mm/s, line width = 0.4 mm, and line-to-line overlapping ratio = 15%.

The respective photomicrographs of shielding test pieces in Figure 2a_1_, b_1_, c_1_, and d_1_ are provided in Figure 2a_2_, b_2_, c_2_, and d_2_, respectively. The different brightness levels of filling lines on the same plane would reflect the smoothness degree of surface structure of each test piece under light microscopy. Accordingly, the presence of lines with different brightness levels on the planes (Figure 2a_2_,b_2_) indicated non-smooth surface test pieces. By comparison, the surface of each test piece in Figure 2c_2_,d_2_ looked smooth because no obvious differences in brightness levels of the planes were observed. On the other hand, the particles marked with red lines (Figure 2d_2_) were identified as boron carbide induced by agglomeration during the experiment instead of the regular forming process. The reason for this had to do with boron carbide ultra-fine powders (G50 = 1.2 µm) possessing high chemical bonding activities (such as hydrogen bonding), which would facilitate agglomeration of particles during experiments.

Melt flow index measurements were performed to evaluate the viscoelastic properties as a necessary characteristic in shaping process operations. Figure 3 presents the change in melt flow index of four materials as a function of temperature under 5000 g weight pressure. In Figure 3, the melt flow indices of all four kinds of materials increased with temperature under the same weight pressure. The matrix PEEK reached the peak melt flow index at about 440 °C. Afterward, the melting index decreased slightly owing to the melting point of PEEK (343 °C). As temperature rose further, PEEK was carbonized into carbides, hindering the melting flow and reducing the melt flow index.

Meanwhile, the nozzle diameter became smaller than the original as excessive carbides were generated and gathered around the nozzle, thereby affecting the line width. Therefore, appropriate temperatures can ensure unobstructed materials without forming new impurities. As the melting point of boron carbide is 2350 °C, it would disperse as solid particles in the matrix PEEK at temperatures below 2350 °C. In Table 1, the volume ratio increased with the mass ratio of boron carbide in the composites. This could exert a negative impact on melt fluidity of carbon fiber. The melting indices of the materials were measured according to GB/T 3682-2000 standard [75]. Note that melting index refers to the fluidity value of thermoplastic materials during processing. According to GB/T 3682-2000 standard, thermoplastic materials were first melt into fluid within 10 min at a predetermined pressure and temperature. The mass (g) flowing through the circular tube with 2.095 mm in diameter was then measured as the melting index. Note that the melting index value was proportional to the material fluidity. In other words, larger values would induce greater fluidity, and vice versa. However, excessively large fluidity values would render the fluid after the melting process more difficult to shape when passing through the nozzle, while materials with extremely low fluidities may lead to clogging in the nozzle, limiting their passage through time.

In Table 1, the corresponding volume ratio increased with boron carbide content owing to the presence of boron carbide particles in the composites. On the other hand, the melting index of each composite decreased with boron carbide content at the same temperature (Figure 3). At load of 5000 g, PEEK melting index of 13 g/10 min, and temperature of 410 °C, PEEK was successfully extruded without carbonization or clogging in the nozzle. Using good fluidity composites, less PEEK carbonization, and a low blocking nozzle rate, the melting indices of 10% wt., 20% wt., and 30% wt. boron carbide PEEK were set to 11 g/10 min at a printing temperature of 425 °C, 10.4 g/10 min at 435 °C, and 9.8 g/10 min at 440 °C, respectively.

Here, 10% boron carbide PEEK composite with wire diameter of 1.75 ± 0.05 mm was used in the experiments (Figure 4).

During processing, a 3D printer with nozzle diameter of 0.4 mm, nozzle temperature of 425 °C, movement speed of 20 mm/s, and line width of 0.4 mm was employed for processing. The corresponding density was calculated under different overlapping ratios of filling lines. Note that overlapping ratios of 5% to 15% induced densities of 98% to 99.2%.

Figure 5a illustrates a light micrograph diagram of the test piece surface at filling line overlapping ratio of 2% (0.008 mm). The filling lines looked clear with paddings in between the lines. In Figure 5b, the surface of test pieces tended to uniform planes with a flat surface at an overlapping ratio of 15% (0.06 mm). Moreover, in Figure 5c, large amounts of overflow occurred at the junction of filling lines and overlapping ratio of 50% (0.2 mm). Excess material accumulated on the surface, leading to an uneven surface. Therefore, the printing of the next layers generated many holes. Furthermore, high temperature accelerated carbonization, thereby inducing more impurities and a large numbers of air holes in printed samples. This, in turn, reduced the quality of test pieces. All test pieces in Figure 5a–c contained 10% wt. boron carbide PEEK.

Test shielding neutron cylinders with 55 mm in diameter are shown in Figure 2c_1_,d_1_, and the PEEK shielding neutron test piece was presented in Figure 2c_1_. For FDM processing of the boron carbide PEEK/PEEK composites with a 0.4 mm diameter nozzle (Table 2), the nozzle speed was set to 20 mm/s and the diameter of print nozzle (line width) was 0.4 mm. Its line-to-line overlapping ratio of PEEK was 10%, and that of PEEK composites was 15%. The respective nozzle temperature of PEEK, 10% boron carbide PEEK, 20% boron carbide PEEK, and 30% boron carbide PEEK was fixed at 410 °C, 425 °C, 435 °C, and 440 °C, respectively.

### 2.3. Analysis and Testing 

#### 2.3.1. Permeability Test and Void Ratio of Boron Carbide PEEK

The density of each test piece was measured by the drainage method, requiring dry test samples without absorbed water. Consequently, water absorption of both PEEK and boron carbide PEEK composites was analyzed. After drying at 150 °C for 10 h, PEEK, 10% wt. boron carbide PEEK, 20% wt. boron carbide PEEK, and 30% wt. boron carbide PEEK were separately immersed in purified water for 10 min followed by drying by filter paper and air stream. The moisture of wire materials before and after soaking was evaluated by a moisture meter for 20 min. Figure 6 displays the water absorption profiles of boron carbide PEEK composites. Soaked and unsoaked specimens demonstrated no significant differences in water content. Thus, the drainage method could feasibly be used to test the densities of PEEK and boron carbide PEEK composite.

The theoretical density of PEEK composites can be calculated by Equation (1). The ratio of actual density of test pieces measured by the drainage method to the theoretical density would present the compactness criteria.
(1)$$\rho =\frac{m}{V_{1}+V_{2}}=\frac{{m\rho }_{1}\rho_{2}}{{m\rho }_{2}+{m\rho }_{1}}=\frac{{m\rho }_{1}\rho_{2}}{{\eta m\rho }_{2}+\left(1-\eta \right){m\rho }_{1}}=\frac{\rho_{1}\rho_{2}}{{\eta \rho }_{2}+\left(1-\eta \right)\rho_{1}}=\frac{\rho_{1}\rho_{2}}{\eta \left(\rho_{1}-\rho_{2}\right)+\rho_{1}}$$
where m is the total material mass after compositing (in g), $\rho_{1}$ is the reinforcing material density (in g/cm^3^), $\rho_{2}$ is the matrix material density (in g/cm^3^), ρ is the composite material density (in g/cm^3^), and η is the mass ratio of composite materials. $V_{1}$ represents the volume of reinforcing material (in cm^3^), and $V_{2}$ is the volume of matrix material (in cm^3^).

With 100% compactness (Table 3), the matrix PEEK was filled with melted powder without the formation of bubbles. The compactness of the composite material decreased with mass of boron carbide owing to slight agglomeration of boron carbide powders during the mixing of dry powders.

#### 2.3.2. Neuron Shield Platform and Shielding Performance Test Method

The samples and center of boron trifluoride detector (BF_3_ detector) were placed on the same central axis and the long counter of the BF_3_ detector was used for counting. The collection and storage of neutron shielding experiments and neutron shielding platform are displayed in Figure 7. The red dot represents the Am-Be source and front surface of test pieces to shield was set 3 cm away from the geometric center of the Am-Be neutron source. The acquisition time was programmed to 30 s at a test voltage of 1600 V. The neutron shielding performance of composite materials was calculated according to Equations (2) and (3):(2)$$E\left(\%\right)=\frac{C_{0}-C_{i}}{C_{0}}\times 100\%$$
(3)$$C_{0}=\frac{n_{0}}{t_{0}},\;C_{i}=\frac{n_{i}}{t_{i}}$$
where *t*_0_ represents the real acquisition time (s) in absence of testing pieces, *t_i_* is the real acquisition time (s) of the *i*th test, *n*_0_ means the number of neutrons detected by the detector in absence of testing pieces, *n_i_* is the number of neutrons detected by the detector in the *i*th test, and E represents the neutron shielding ratio in the form of percentage. Moreover, *C*_0_ and *C_i_* are the numbers of neutrons detected within 1 s in non-test pieces and test pieces, respectively. *E*% denotes the shielding efficiency.

## 3. Results and Discussion

### 3.1. Mechanical Properties

In Figure 8, the test piece was made by FDM according to the standards GB/T 1040.2-2006 [76] and GB/T 9341-2008 [77]. Note that GB/T 1040.2-2006 standard represented a state-mandated test condition used to determine the tensile properties of molding and extruded plastics through specifying the shape, size, and number and test procedures of test pieces. In this standard (Figure 8a), the speed of the tester used for measuring the tensile properties was set to 1 mm/min, except for dumbbell-shaped test pieces. Besides, this standard was equivalent to ISO 527-2:1993.GB/T 9341-2008 utilized for the determination of bending properties of plastics, issued by the General Administration of Quality Supervision, Inspection, and Quarantine of the People’s Republic of China in 2008 then implemented on 1 April 2009. This standard was equivalent to ISO 178:2001 and could be applied to thermoplastic composites as it specified the dimensions, numbers, and procedures of test molding pieces. Figure 8b displayed a test piece cuboid with 80 mm in length, 10 mm in width, and 4 mm in height as tested by GB/T 9341-2008 standard at a speed of 1 mm/min.

The tensile strength of PEEK (before heat treatment) was estimated to be 59.29 MPa (Figure 9a). As the mass of boron carbide was enhanced, tensile strength gradually declined. In other words, the increase in mass of boron carbide led to a decrease in the compactness of the composite materials, thereby influencing the printing performances. To improve the mechanical properties, the test pieces were first fixed and then incubated at 300 °C for 2 h. The tensile strengths of both matrix PEEK and PEEK composites increased by 40%–50%.

The flexural strength in PEEK composites reached a maximum at boron carbide content of 10% (Figure 9b). To facilitate the agglomeration of ultrafine powders during mixing, the size of boron carbide particle was set to G50 = 1.2 µm. The boron carbide in PEEK occupied 5.62% vol. in 10% wt. boron carbide PEEK, 11.81% vol. in 20% wt., and 18.67% vol. in 30% wt. During the printing process at a nozzle temperature of 440 °C, boron carbide was kept in its original particle form without melting. At boron carbide content of 0–10% wt., PEEK could tightly pull the boron carbide. At contents exceeding 10%, the agglomeration of boron carbide increased with boron carbide content and both proportion of PEEK and corresponding flexural strength decreased. In Table 2, the porosity of PEEK test pieces printed by FDM reached 99.54%, and those of 10% wt., 20% wt., and 30% wt. boron carbide PEEK attained 99.19%, 98.74%, and 98.27%, respectively. At boron carbide contents above 10% wt., the porosity of test pieces rose, leading to a decline in flexural strength. At contents exceeding 30%, the flexural strength reduced rapidly as the agglomeration of boron carbide at low contents induced little effect on the materials. That is to say that large specific gravity PEEK completely coated boron carbide at contents below 10% after melting at 440 °C, but changed as boron carbide content was further enhanced. Above 30%, PEEK did not fully immerse in boron carbide after melting, thereby diminishing the flexural strength of obtained composite materials. After heat treatment, the flexural strength increased from 65% to 78%. Hence, heat treatment could improve the mechanical properties of materials.

### 3.2. Heat Deformation Temperature

Test pieces were made following the standard GB/T 1634.2-2004 used for temperature determination of plastics under load application [78]. This standard was issued by the General Administration of Quality Supervision, Inspection, and Quarantine of the People’s Republic of China and the Standardization Administration of China on 15 March 2004 and then implemented on 1 December 2004. GB/T 1634 specifies the temperature at which test disturbance of an 80 mm × 10 mm × 4 mm plastic cuboid piece reaches the standard deflection value under 0.45 MPa load with the increase in test temperatures. The principle states that, under constant three-point bending load, the sample should produce a specific bending stress in relevant parts of GB/T 1634.2-2004. Besides, the standard deflection temperature corresponding to the specified bending strain increment should be measured under constant temperature increasing conditions. Table 4 shows the results of heat deformation temperature tests of polyethylene and PEEK/PEEK composite materials. The test results of injection-molding polyethylene test pieces provided in the literature were used for comparison [79].

Figure 10 presented the heat deformation temperature profiles of PEEK and boron carbide PEEK composites at 0.45 MPa. The matrix PEEK displayed high heat deformation temperature. As boron carbide content increased, the heat deformation temperature gradually rose owing to the small thermal conductivity of boron carbide with little effect on the matrix material. Using GB/ T 1634.2-2004 standard, the sample produced specific bending stress in relevant parts of GB/T 1634 under constant three-point bending load (0.45 MPa). Besides, the standard deflection temperature corresponding to specific bending strain increment was measured under constant temperature increase conditions. As temperature rose to 300 °C, but before reaching the melting point of PEEK (343 °C), PEEK started to re-crystallize [80], leading to the formation of solid glue tightly holding boron carbide. At this point, boron carbide acted as supporting material, which led to increased bending strength. Thus, the thermal deformation temperature of 30% boron carbide PEEK was higher than that of 20% boron carbide PEEK, but this did not mean that the thermal deformation temperature rose with boron carbide content. As boron carbide content was further enhanced, the corresponding PEEK content decreased owing to the reduction in heat deformation temperature, where decreased PEEK could no longer hold boron carbide. At this point, the incrementing trend of heat deformation temperature was only suitable for boron carbide contents below 30%. In Table 3, the heat deformation temperature of PEEK/PEEK composites was four- to fivefold superior to that of PE. Thus, PEEK/PEEK composites possessed better high-temperature resistance.

### 3.3. Neutron Shielding Performance 

#### 3.3.1. Mono-Layer Shielding Body Structure

FDM was employed to prepare solid filled cylindrical mono-layer shielding test pieces (55 mm in diameter) of PEEK, 10% wt. boron carbide PEEK, 20% wt. boron carbide PEEK, and 30% wt. boron carbide PEEK. Test pieces with the same material, but various thicknesses, namely, 20 mm, 40 mm, 60 mm, 80 mm, and 100 mm, were used for experimentation.

The fast neutron shielding process in neutron science is often accomplished in two steps: slowing down fast neutrons via H followed by absorption of slow neutrons into boron carbide. For the Am-Be source used in experiments (Figure 11), the majority of the shielding surface near the direct radiation source was made of fast neutrons. Using boron carbide PEEK as new shielding material, the matrix PEEK containing H slowed down the neutrons, thereby reinforcing boron carbide as a neutron absorber. To ensure the same neutron radiation intensity, the test pieces were placed 3 cm away from the Am-Be source. Boron carbide occupied 11.81% vol. in 20% wt. boron carbide PEEK, while PEEK took up 88.19% vol. For 30% wt. boron carbide PEEK, boron carbide occupied 18.67% and PEEK took up 81.33% vol. The shielding ratio of PEEK increased with the thickness of test pieces. Using the same thickness, the shielding rate of PEEK composites rose with boron carbide content. Boron carbide PEEK composites with the same content showed an enhanced neutron shielding rate with the thickness of test pieces. Thus, PEEK and boron carbide PEEK composites were good neutron-absorbing materials. For PEEK composites with 30% wt. boron carbide content and shielding body thickness of 100 mm, the neutron shielding rate reached 88.24%. At shielding body thicknesses exceeding 20 mm, the neutron shielding ratio changed exponentially with boron carbide content. At 5% wt.–10% wt. boron carbide, the slope of the curve changed greatly, indicating a significant shielding effect. Above 10%, the slope tended to a plateau, meaning a strong shielding effect of composites in the presence of 10% wt. boron carbide.

It was encouraging to notice that 20% boron carbide PEEK was optimal for 20 mm shielding test pieces. In other words, the best content of boron carbide was around 20% wt. for a shielding test piece thickness of 20 mm. For 20 mm shielding testing body with 30% wt. boron carbide PEEK, H content was less than 20% wt. After passing through a 20 mm 30% wt. boron carbide PEEK shielding body, the fast neutrons can no longer slow down owing to the presence of less H element than in a 20% wt. boron carbide PEEK shielding body. Therefore, the fast neutrons would escape and cannot be captured by boron carbide, leading to a decrease in shielding efficiency. For 0% wt. to 20% wt. contents, the shielding ratio increased as H content reduced and boron carbide content rose. Thus, H can effectively slow down the speed of fast neutrons. However, the neutrons largely escaped at low boron carbide contents and boron carbide was not enough to absorb weakened fast neutrons. Therefore, the optimal shielding plate with reasonable PEEK (H) and boron carbide content distribution might be designed under given thickness to ensure the best shielding effect and maximum material utilization ratio. Besides, the special shielding process of neutron radiation extended the application fields of FDM.

The direct relationship between fast neutron shielding performance of boron carbide PEEK composites and main influencing factors (boron carbide content, shielding piece thickness, and the activity of the radiation source) were obtained through analysis of the data. To achieve the best shielding effect in terms of materials and structural design under space constraints, the best boron carbide content and optimal thickness of shielding body were simulated.

#### 3.3.2. Double-Layer Shielding Body Structure

Figure 12 shows the respective double-layer shielding body of two composite materials with the same thickness (80 mm). The shielding rate of the test piece with (30% + 20%) structure was similar to that of (20% + 30%) structure. In addition, (10% + 30%) structure displayed slightly better shielding performance than (30% + 10%). On the other hand, (0% + 30 %) structure did significantly much better than (30% + 0%), (30% + 10%), and (30% + 20%) structures. 

The reason for that had to do with the significant role of H and boron carbide in shielding fast neutrons. The shielding efficiency was related to distribution of H and boron carbide played in the test pieces. The detailed reasons were provided in Figure 13. Using the same active Am-Be source, A, B, C, and D were employed in shielding test pieces. A and B were put in the middle parts of test bodies at the same height, while C and D were placed on the upper surface of test bodies at the same linear distance from the source space. Supposing (A and B) and (C and D) on the same positions, (C and D) would share the same radiation intensity under the same source and (A and B) should share the same neutron radiation intensity. The passage of neutrons through a shield plate of 80 mm in thickness will induce better shielding neutron effect of the (30% + 30%) structure when compared with (30% + 20%), as (C and D) were close to the source and most were fast neutrons. At (A and B), the fast neutrons slowed down to become slow neutrons. After the passage of slow neutrons through the next 40 mm shield body, the 30% wt. boron carbide PEEK absorbed more than 20% wt. boron carbide PEEK. Moreover, 30% wt. boron carbide PEEK contained more boron carbide than 20% wt. boron carbide PEEK. For 80 mm (20% + 30%) and 80 mm (30% + 20%) structures, the passage of neutrons through the first 40 mm thick layer yielded lower shielding efficiency for 20% wt. boron carbide PEEK when compared with 30% wt. boron carbide PEEK. However, compensation occurred when neutrons passed through the next 40 mm thick layer as 30% wt. boron carbide PEEK could absorb more neutrons than 20% wt. boron carbide PEEK. Thus, 80 mm (20% + 30%) and 80 mm (30% + 20%) structures presented the same shielding efficiency. Meanwhile, the shielding efficiency of 80 mm (30% + 30%) was similar to that of 80 mm (10% + 30%) in terms of neutron shielding efficiency of the double-layer shielding body structure (Figure 12). Both materials can shield 82.28% of neutrons. However, 80 mm (0% + 30%) outperformed both 80 mm (30% + 0%) and 80 mm (10% + 30%) in terms of shielding efficiency. As a result, 80 mm (0% + 30%) structure was superior to both 80 mm (30% + 30%) and 80 mm (10% + 30%) structures in terms of material utilization (reduced cost), shielding body weight, and shielding efficiency.

As neutron shielding is complex, the next stage was devoted to calculating the best ratio distribution of materials and designing the optimal structure based on material distribution by nuclear shielding simulation. However, injection molding and molding of complex geometry could only slightly cope with different gradient contents of shielding protection. Thus, FDM was adopted as it does not require expensive molds and limitation of the geometrical structure of shielding bodies. FDM often excels in complex structure manufacturing with broader applications. Despite this, the method achieved good integration prospects in design and manufacturing under optimal shielding performance in terms of shortened processing time, reduced material waste, and minimized production costs.

## 4. Conclusions

A new type of Boron Carbide PEEK composites for neutron radiation shielding was prepared by FDM processing. The resulting materials were tested and their properties were evaluated. The following conclusions could be drawn:(1)To reduce printing failures and improve compactness of test pieces with reduced defects, the printing temperature of novel boron carbide PEEK composites suitable for FDM was changed from 420 °C to 440 ° C and overlapping rate of filling lines varied from 5% to 15%.(2)Both matrix material PEEK and boron carbide PEEK composite recorded heat deformation temperature around 300 °C under load of 0.45 MPa. This solved issues brought by excessive low deformation temperature of polyethylene composite and led to excellent mechanical properties of boron carbide PEEK composite after heat treatment.(3)The shielding tests showed the novel boron carbide PEEK materials to possess excellent fast neutron shielding performances. Shielding pieces with 100 mm in thickness and 30% boron carbide content could absorb up to 88.24% neutrons. For thicknesses from 40 mm to 100 mm, the neutron shielding ratio changed exponentially with the content of boron carbide.(4)Neutron shielding composite materials and shielding structures were then produced according to composition and structure to yield nuclear shielding materials with multiple shielding characteristics and shielding/structure integration. In sum, FDM could be used for the design and production of variable-structure neutron shielding bodies with excellent mechanical loading capacities and shielding performances. Such structures should be characterized by small size, lightweight, less consumables, and low cost.

## Figures and Tables

**Figure 1 materials-13-02314-f001:**
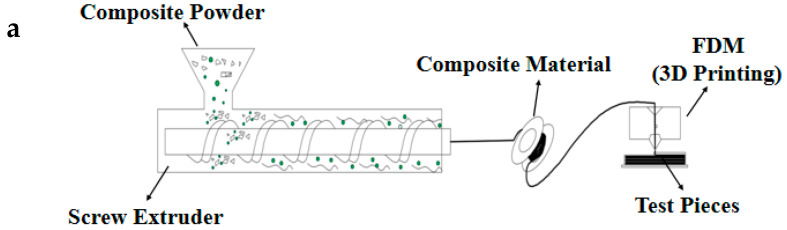
(**a**) Preparation of new shielding materials, (**b**) fused deposition modeling (FDM) special material of boron carbide polyether-ether-ketone (PEEK) composites, and (**c**) FDM processing flow diagram.

**Figure 2 materials-13-02314-f002:**
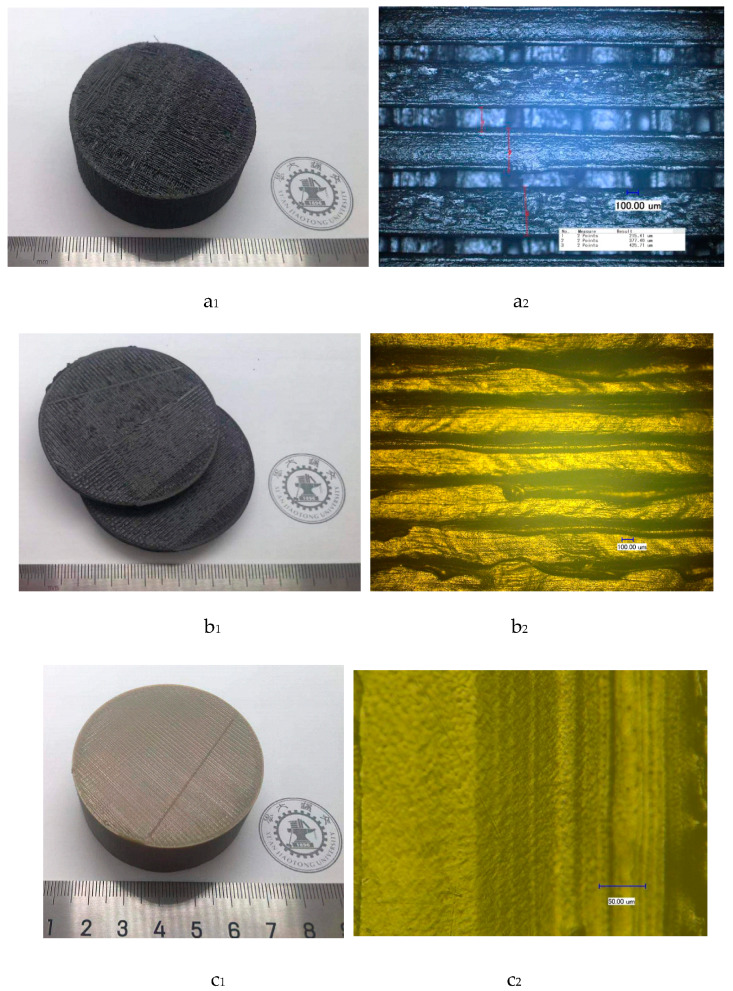
Surface structure diagrams of 3D printing (FDM) test pieces. (**a_1_**) and (**b_1_**) Unqualified test pieces. (**c_1_**) and (**d_1_**) Qualified test pieces. (**a_2_**), (**b_2_**), (**c_2_**), and (**d_2_**) Light microscopic images corresponding to (**a_1_**), (**b_1_**), (**c_1_**), and (**d_1_**), respectively.

**Figure 3 materials-13-02314-f003:**
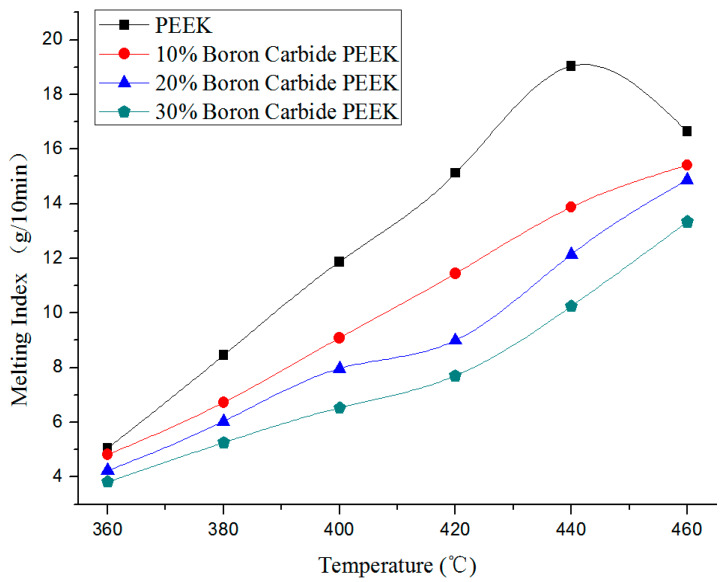
Change in melting index of boron carbide PEEK composites as a function of temperatures (5000 g).

**Figure 4 materials-13-02314-f004:**
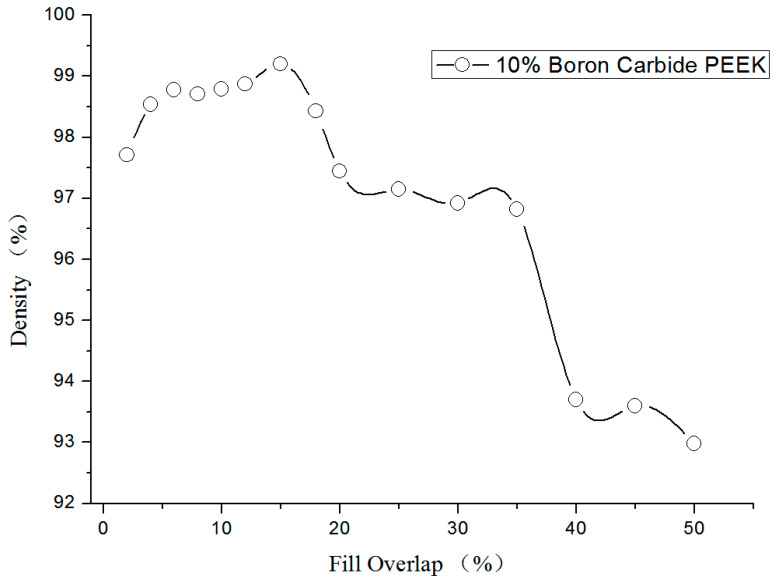
Relationship between the overlapping ratio of filling lines with density.

**Figure 5 materials-13-02314-f005:**
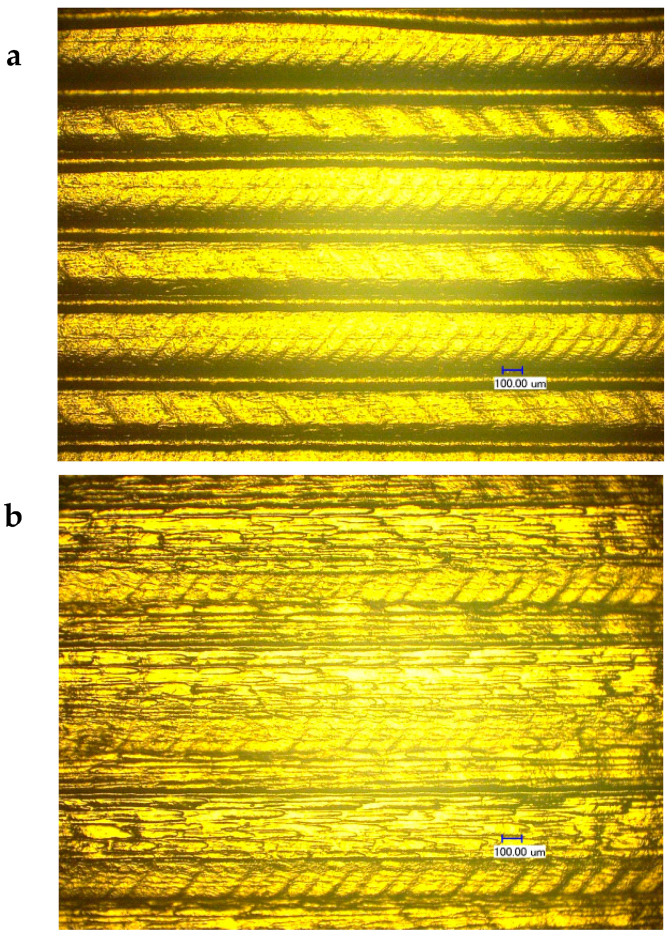
Light micrograph diagram of test piece surface. (**a**) Line-to-line overlapping ratio = 2%; (**b**) Line-to-line overlapping ratio = 15%; (**c**) Line-to-line overlapping ratio = 50%.

**Figure 6 materials-13-02314-f006:**
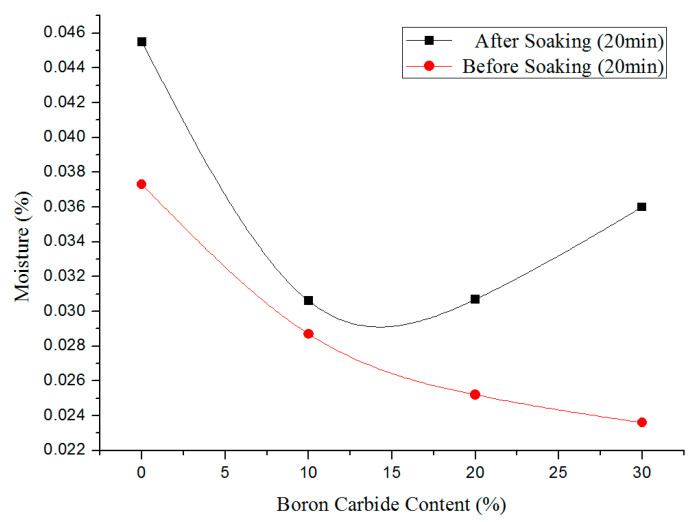
Change in water content of boron carbide PEEK composite before and after soaking in water.

**Figure 7 materials-13-02314-f007:**
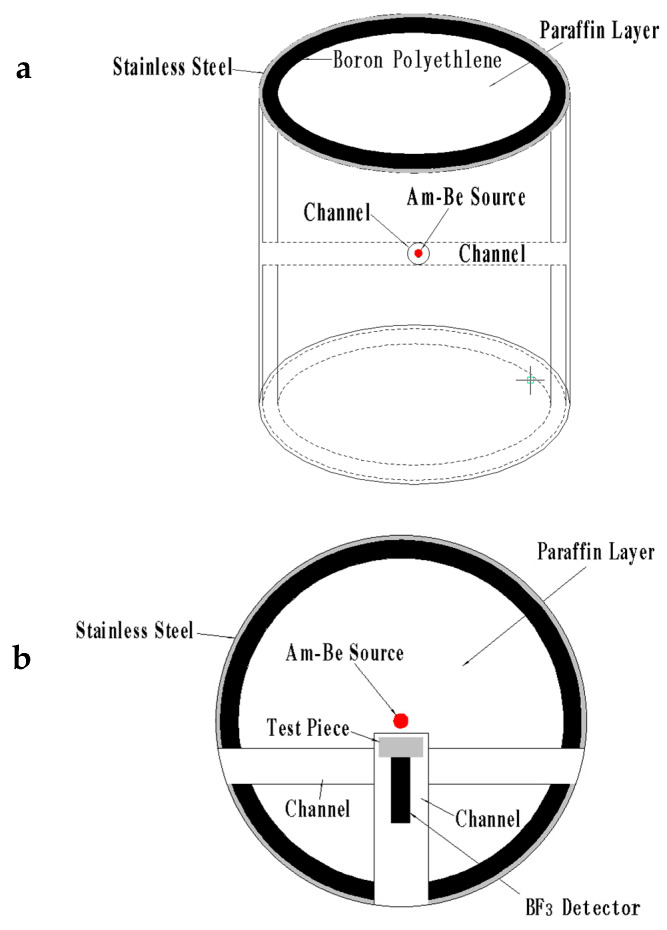
Structure of neutron source storage and experiment platform: (**a**) front view and (**b**) top view.

**Figure 8 materials-13-02314-f008:**
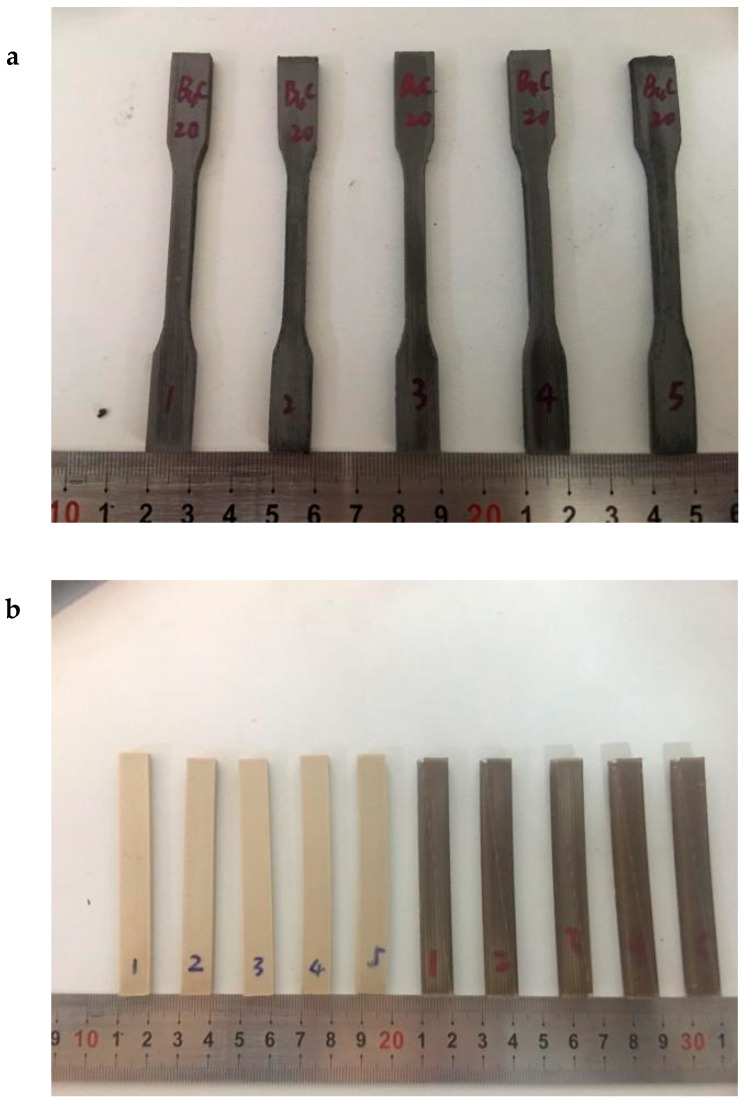
Mechanical properties of test pieces: (**a**) tensile strength test pieces and (**b**) flexural strength test pieces.

**Figure 9 materials-13-02314-f009:**
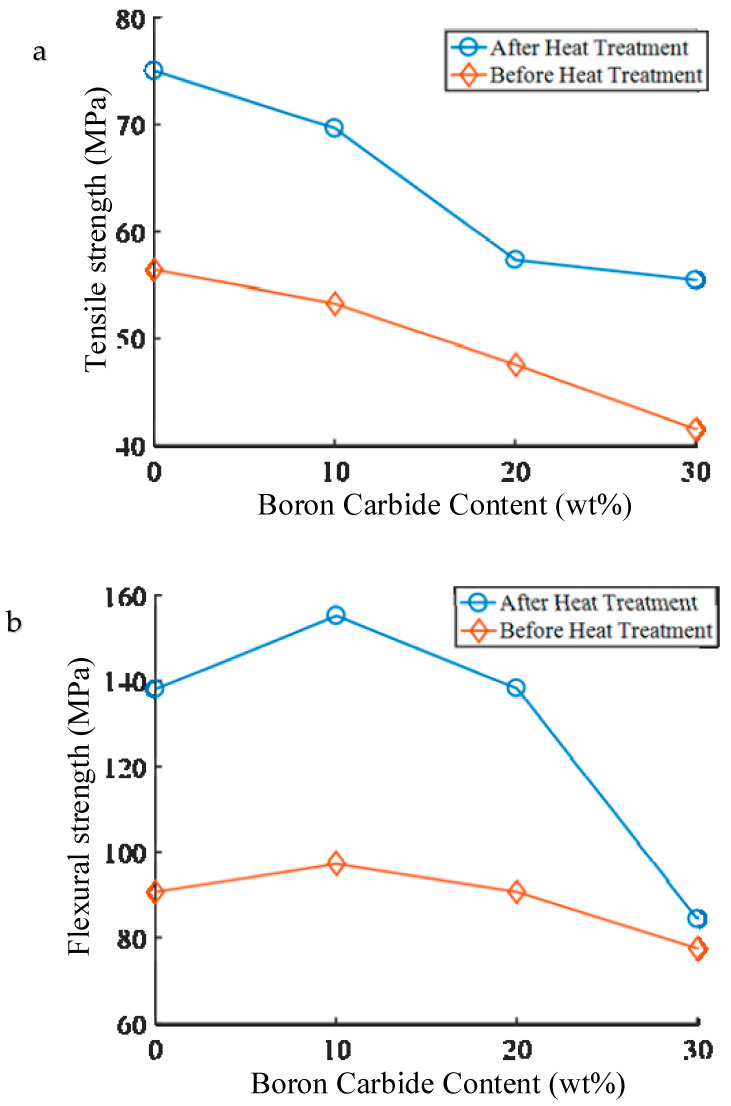
Tensile and flexural strengths of new composite shielding materials. (**a**) Test result of tensile experiment and (**b**) test data of flexural experiments.

**Figure 10 materials-13-02314-f010:**
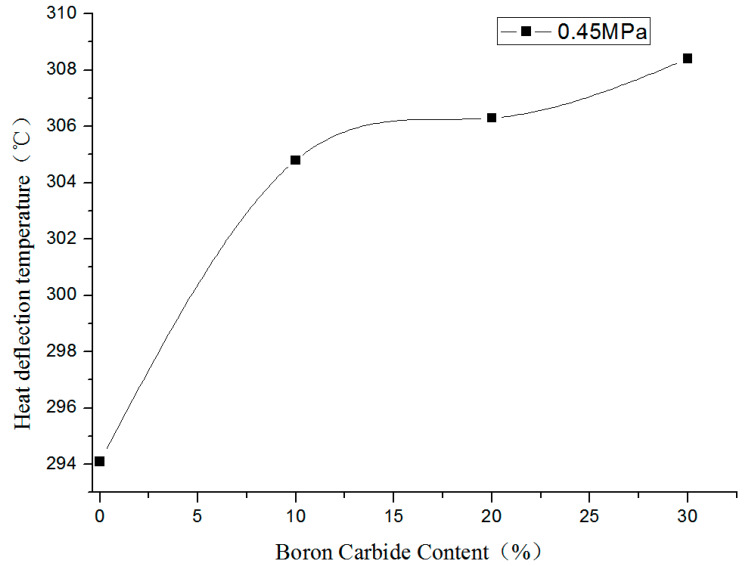
Changes in heat deformation temperature of boron carbide PEEK composites.

**Figure 11 materials-13-02314-f011:**
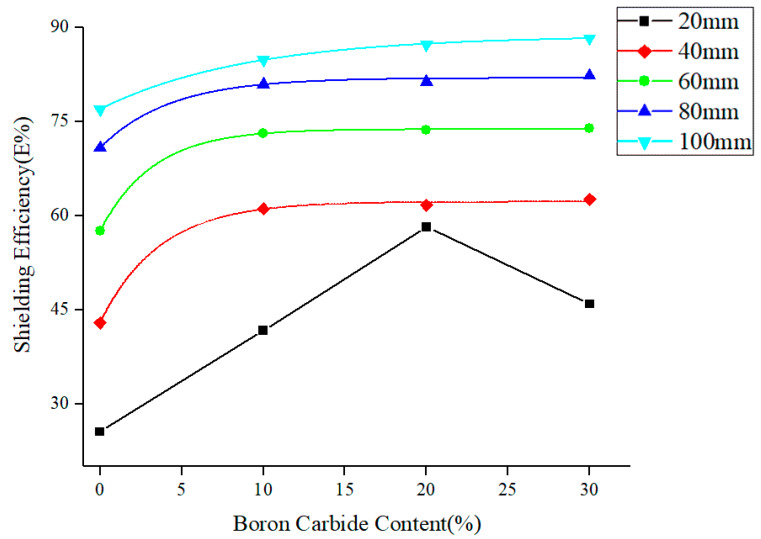
Shielding performance of PEEK/boron carbide PEEK on the Am-Be neutron source.

**Figure 12 materials-13-02314-f012:**
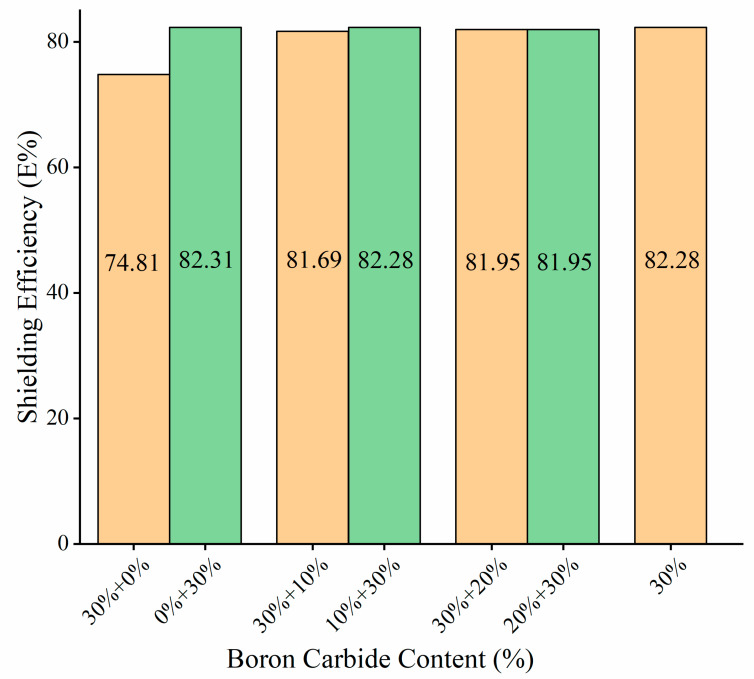
Effect of shielding performance of PEEK/boron carbide PEEK with special structure on the Am-Be neutron source. Note that the thickness of test pieces is 80 mm. “30% + 0%” means double-layer shielding body during testing. 30% wt. boron carbide PEEK material with 40 mm in thickness is close to the detector. Neat PEEK material thickness is 40 mm.

**Figure 13 materials-13-02314-f013:**
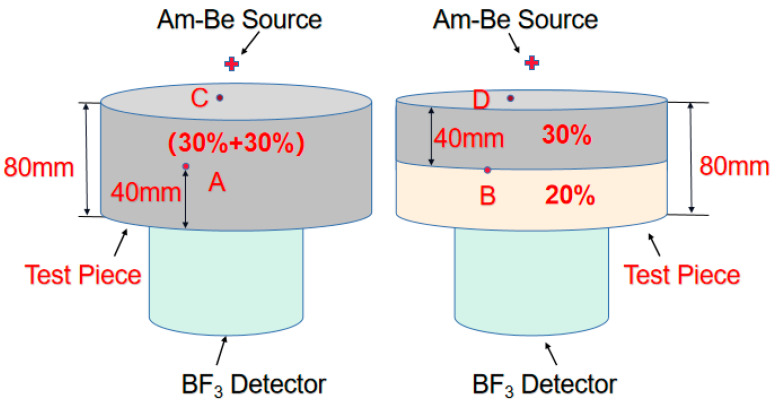
Neutron shielding experimental platform of shielding test body with double-layer structure.

**Table 1 materials-13-02314-t001:** Fused deposition modeling (FDM) temperature parameters of boron carbide polyether-ether-ketone (PEEK).

Composites Name	Volume Ratio of Boron Carbide (vol.%)	Melting Index (g/10 min)	Printing temperature (°C)
PEEK	0	13.00	410 °C
10% wt. Boron Carbide PEEK	5.62	11.00	425 °C
20% wt. Boron Carbide PEEK	11.81	10.40	435 °C
30% wt. Boron Carbide PEEK	18.67	9.80	440 °C

**Table 2 materials-13-02314-t002:** FDM processing parameters of boron carbide PEEK.

Composites Name	Nozzle Temperature (°C)	Nozzle Speed (mm/s)	Diameter of Print Nozzle (mm)	Line Width (mm)	Line-to-Line Overlapping Ratio
PEEK	410	20	0.4	0.4	10%
10% wt. Boron Carbide PEEK	425	20	0.4	0.4	15%
20% wt. Boron Carbide PEEK	435	20	0.4	0.4	15%
30% wt. Boron Carbide PEEK	440	20	0.4	0.4	15%

**Table 3 materials-13-02314-t003:** Density and compactness of PEEK and boron carbide PEEK composites.

Composites Name	Actual Wire Density (g/cm^3^)	Theoretical Wire Density (g/cm^3^)	Test Piece Density (g/cm^3^)	Wire Compactness	Test Piece Compactness
PEEK	1.300	1.30(VICTREX)	1.294	100%	99.54%
10% wt. Boron Carbide PEEK	1.365	1.366	1.354	99.93%	99.19%
20% wt. Boron Carbide PEEK	1.429	1.439	1.411	99.31%	98.74%
30% wt. Boron Carbide PEEK	1.499	1.520	1.473	98.62%	98.27%

**Table 4 materials-13-02314-t004:** Heat deformation temperature and tensile strength of PEEK/boron carbide PEEK composites/polyethylene.

Materials	Heat Deformation Temperature (GB/T 1634.2-2004)	Tensile Strength (GB/T 1040.2-2006)
VICTREX PEEK (FDM 3D Printing)	294.1 °C	75.08 MPa
10% wt. Boron Carbide PEEK (FDM 3D Printing)	304.8 °C	69.68 MPa
20% wt. Boron Carbide PEEK (FDM 3D Printing)	306.3 °C	57.40 MPa
30% wt. Boron Carbide PEEK (FDM 3D Printing)	308.4 °C	55.45 MPa
High-density polyethylene 2911 produced by PetroChina Fushun Petrochemical Company (Injection Molding) [79]	69.6 °C	27.10 MPa
High-density polyethylene DMDA-8008 produced by PetroChina Dushanzi Petrochemical Company (Injection Molding) [79]	73.7 °C	24.80 MPa
High-density polyethylene C430 produced by Korea Samsung Total) (Injection Molding) [79]	63.6 °C	24.70 MPa
High-density polyethylene ME3500 produced by Korea LG Group (Injection Molding) [79]	67.7 °C	24.50 MPa

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
