# Peer review of "Neutron Shielding Performance of 3D-Printed Boron Carbide PEEK Composites"

_materials, 2020, doi:10.3390/ma13102314_

Round 1

Reviewer 1 Report

The present paper deals with the testing and properties study of a new type of Boron Carbide PEEK composites for neutron radiation shielding prepared by FDM processing.

I find that the scientific problem discussed in this paper is original and of scientific interest but I have some major criticisms:

- At page 2, line 65, make explicit the acronym FDM that is “Fused Deposition Modeling”;

- At page 4, line 118, explain better in which way the cited printing temperatures were chosen;

- At page 5, move “Figure 4” from line 123 to line 126;

- At page 6, “Figure 2e” is cited at lines 142 and 145 but Figure 2e is not reported in the paper; please insert it;

- At page 6, add a reference about Eq. (1);

- Explain better at page 11 the design of the double-layer shielding bodies;

- Moreover, in my opinion the authors should discuss in deeper detail in the “Results and discussion” section, the advantages of the presented Boron Carbide PEEK composites.

I ask also the authors to check some English grammar, syntax and typing along the whole paper.

Reviewer 2 Report

Novel Boron Carbide Polyether-Ether-Ketone composites are presented as shielding pieces manufactured through FDM. Heat resistance, mechanical properties and neutron shielding characteristics were evaluated and reported.

After Heat Treatment bending and tensile strength are reported as increased.

Neutron shielding rate is reported to exhibit exponential change as function of thickness and boron carbide content. Please, see my comment below on this particular point. In the conclusions, a suggested optimal amount of boron carbide is not presented as suggested by the abstract.

Specific comments and suggestions:

Line 5: typo in "Aerospace"

Line 25: what types of "rays"?

Introduction section:
Line 31: "strongly penetrate" Please, this sentence seems incomplete.
Line 33: missing verb? ... " but (?) not convenient..."

Line 42: do authors mean: high boron-content or high content OF boron?

Experimental Section:

Quality of Figure 1 needs improvement, larger font size, typo "screw extruder".

Fig 2: Please, can you add some explanation on text of what we are seeing in a2, b2,c2,d2? Are these light micrographs as those shown in Fig. 5?

Line 118 typo: "was were"?

Line 142 and 145 : there is no Fig 2e. Line 143 repeats reference to Fig 2d.
Line 162-164: is that the definition of compactness?

Line 165: ρ1 seems to not be mass, it has units of density!

Line 173: sentence needs rewriting.
Line 175: missing word? The red dot?

Section 2.3.2:

Eqn. 2 and 3: meaning of Co and Ci is missing from the text

Maybe useful to say that measure E% values are reported on this paper in section 3.3? E% seems to be the shielding efficiency mentioned in Figure 10.

Fig 7: Parafin layer: not clear its location. Can you please show it on the perspective image in the left hand side of figure 7?

Section 3, Results and Discussion:

Line 187: Can standards GB/T 1040 and 9341 be referenced or at least their titles and/or brief description be included in the text?
Line 207: similar request for GB/T 1634.

Fig 8: please improve quality of the image. Larger font size is needed.
No explanation of how the experiment is performed or used equipment, therefore the references for GB/T are needed.

Line 200 to 202: Confusing explanation. it is not clear the writer's supporting argument for the decrease of bending strength for boron content from 10% to 30%.

Table 2: tensile strength values 75.08, 69.68, 57.40 and 55.45 MPa were also measured following GB/T 1040? According to figure 8, these are values after heat treatment. It would help the reader to write, for instance in line 197 that these values are shown in table 2.

Line 216-217: "High 216 boron carbide content had supporting role" what does this mean? Would this explain the increase in the thermal deformation temperature?

Fig 9: Needs consistency in the name of properties: heat deflection or thermal deformation temperature?

Fig 10: Any explanation for the apparently higher than expected shielding observed for the 20% Boron Carbide 20mm thick sample?
Could this be an experimental error?  In lines 21 and 22, and in conclusion #3, the authors mention that "the neutron shielding rates exhibited exponential change as a function of thickness of shielding body and boron carbide content", however, this seems not to be the case for 20% boron carbide PEEK, where E% is not an exponential function of thickness.

Lines 236 to 240: The explanation is not clear "Before reaching the limit value, shielding effect increased with content then declined afterward because both boron carbide and hydrogen contents in shielding pieces with equal thickness affected the shielding. In particular, fast neutrons eventually slowed down thermal neutrons after elastic collision with hydrogen nuclei then absorbed by boron carbide after certain diffusion length."

In fact, in Figure 11, 30%+30% shows a higher value of E% than 20%+30% and 30%+20%, which seems to contradict figure 10 regarding the E% for 20mm-thick 20% B content being higher that E% for 30% B content.

Conclusions:
Should this be section 4, not 3?

References:
Maybe renumbering of references 8, 9, 10, 11 and 12 is needed. They do not appear in order in the text.

Round 2

Reviewer 1 Report

I find the authors have sufficiently fulfilled all the requirements I pointed out in my previous report, therefore I recommend the publication of the present paper in Materials.

Author Response

Dear Reviewer,

 Thank you for your review!

 Kind regards,

                                   Yin Wu,Yi Cao,Ying Wu,Dichen Li

                                        4,May,2020

Reviewer 2 Report

Very good improvement of the manuscript. Thank you!

Please, let me include 6 main points with questions on 2 sections and few minor corrections. I have also attached a pdf file including 2 image captures from the manuscript.

Regarding section 2.2 Shielding process:

Firstly, let me acknowledge the good job the authors do in explaining what are learning from Figure 5 (lines 151 to 158) when different overlapping ratio are used in each case 5a, 5b and 5c.

I appreciate the authors have included an explanation of what we are seeing in these micrograph images in Figure 5.   

1) Can one assume figure 5 shows surface micrographs of test pieces, all of them with same composition (10% boron carbide PEEK composite, from lines 146 to 150)?

Once again, I appreciate the authors present a comprehensive analysis on the light micrographs of figure 5 a, b and c for different filling line overlapping (2%, 15% and 50%) and consequent change in the surface patterns: from lines with paddings in between to uniform planes to uneven surface due to uneven overflow and excess material.

Now, regarding what we learn from figure 2, where the authors want to show differences in the 3D printing process of different pieces.

2) What is different (and what is the same) between cases a, b, c, d in Fig. 2? Please centralize this somewhere in this section. Part (but not all) of this information is spread over and therefore difficult to trace.

 Let me explain. At first glance, it seems that composition of the 4 test pieces would be the same, and only printing process parameters, like nozzle speed, diameter (or width of filling lines) and overlap between lines are different.  The parameters and compositions are not clear stated in the text.

Lines 103 to 108 give a qualitative description of differences in 3D printing process parameters. Nothing is said about 2a1, therefore one would assume that everything said for 2b, 2c and 2d is qualitatively relative to case 2a.

This is, from these lines one can understand that possibly:

  • 2b was done with smaller filling than 2a.
  • 2d was done with larger overlap ratio that 2a
  • 2c seems to indicate "increase in moving resistance and delay in movement trajectory", but the reader does not know what parameter changed between 2a, 2b and 2c that changed the nozzle plane and consequent misalignment.

Then, later in lines 119-120 we learn that case 2a had excessive carbides generated and gathered around the nozzle.

However, only after arriving to line 151, is where the reader learns that %BC is different too. Line 89 was already suggesting composition is different, but again, this is not explicit in the text when figure 2 is presented. It is very difficult for the reader to get the information on differences (composition and 3D printing parameters) between cases shown in Figure 2.

Only when arriving to lines 163-168 is where the reader learns that case 2c is PEEK with no boron carbide, and case 2d corresponds to 10% boron carbide PEEK composite. What about 2a and 2b? In this paragraph authors present quantitative information of printing parameters.

Do the values listed in lines 164 and 165 correspond to case 2c? ... what about 2a, 2b and 2c? 

  • nozzle temperature =410°C,
  • nozzle speed= 20mm/s,
  • Is not clear if diameter of print nozzle=0.4 mm, and line width=0.4mm (is this only for case 2c?)
  • Line-to-line overlapping ratio was 10% (0.04mm)

There is no clear information on cases 2a1 and 2b1. We only know that they are "unqualified test pieces". Meaning of “Qualified” versus “Unqualified” is not explained in the text.

3) Text on what we learn from figures 2a2, 2b2, 2c2 and 2d2 is not included on the manuscript.

The reader can see patterns evolving from linear and parallel interfaces, to curvy ones to circular ones. It is clear on 2d2, that the line pattern from filling lines (2a2, 2b2 and 2c2) is not present anymore. Particles are marked with red lines. Are these carbide particles that have formed?

For 2d1 it is mentioned that material has overflowed after melting, but there are no comments pointing towards what is seen in figures 2a2, 2b2, 2c2, and 2d2.

Regarding section 3.3 neutron Shielding Performance.

The addition of extended explanation in the text of section 3.3 is very helpful. I think this add a lot of value to the manuscript. Excellent!

Lines 318-324: "...best content of boron carbide was around 20%wt for shielding test...After passing through 20 mm 30%wt Boron Carbide PEEK shielding body, the fast neutrons cannot slow down due to the presence of less H element than in 20%wt Boron carbide PEEK shielding body. Therefore, the fast neutrons would escape and cannot be captured by Boron Carbide, leading to decreased shielding efficiency". This helps understanding Figure 11 (please see attached pdf file).

Lines 358-359:"After slow neutrons had passed through the next 20 mm shield body, the 30%wt Boron Carbide PEEK would absorb more slow neutrons than 20%wt Boron Carbide PEEK." The addition of figure 13 is also very useful to understand results shown in figure 12 (please see attached pdf file).

Let me ask few more questions please:

4) When reading lines (321-324) and lines (358-359), they seem to convey opposite messages:

"E% of 20%BC PEEK is larger than E% of 30%BC PEEK”, and "E% of 20%BC PEEK is lower than E% for 30%BC PEEK". Could you clarify?

5) Or... would the message be instead that "E% for fast neutron 20%BC PEEK is larger than E% for 30%BC PEEK, and "E% for slow neutron 20%BC PEEK is lower than E% for 30%BC PEEK"? If so, the following question arises:

6) What are the input parameters used for the simulation of the shielding efficiency show in figure 12? Is the measured Shielding Efficiency E% data shown in figure 11 used in the input of this simulation?

If so, then it is surprising to see that 20%+30% and 30%+20% simulated E% have same value and lower that 30%+30%.

Few new minor corrections:

  • Line 101 and 102: "(d2) Light microscopic images of corresponding to (a1), (b1), (c1), and (d1), respectively." Please, clarify. Either one or more words are missing after "of" or "of" needs to be removed.
  • Line 164: “shielding neutron test pieces are presented in Figure 2c1”. Figure 2c1 shows only one test piece.
  • Line 221: Figure 8b shows test pieces, not results.
